# Tolerance of Oncogene-Induced Replication Stress: A Fuel for Genomic Instability

**DOI:** 10.3390/cancers16203507

**Published:** 2024-10-17

**Authors:** Taichi Igarashi, Kimiyoshi Yano, Syoju Endo, Bunsyo Shiotani

**Affiliations:** 1Laboratory of Genome Stress Signaling, National Cancer Center Research Institute, Chuo-ku, Tokyo 104-0045, Japan; taichi-1541@outlook.jp (T.I.); kiyano@ncc.go.jp (K.Y.); shend2@ncc.go.jp (S.E.); 2Department of Biosciences, School of Science, Kitasato University, Minami-ku, Sagamihara-city, Kanagawa 252-0373, Japan; 3Department of NCC Cancer Science, Division of Integrative Molecular Biomedicine, Biomedical Sciences and Engineering, Graduate School of Medical and Dental Sciences, Tokyo Medical and Dental University, Bunkyo-ku, Tokyo 113-8510, Japan; 4Department of Genome Stress Signaling, Institute of Medical Science, Tokyo Medical University, Shinjuku-ku, Tokyo 160-0023, Japan

**Keywords:** oncogene, RAS, cyclin E, MYC, DNA replication, replication stress tolerance

## Abstract

**Simple Summary:**

When oncogenes, which are genes that can cause cancer, become active, they disrupt normal cell processes, especially DNA replication. This disruption is known as replication stress (RS), in which the DNA copying process is stalled or damaged. To prevent cancer, cells usually have a defense system called the DNA damage response (DDR), which can prevent damaged cells from turning into cancer. However, some cells manage to survive this stress by developing replication stress tolerance (RST). These cells can continue to grow, leading to genomic instability (GIN), which is a key feature of cancer. GIN causes various genetic changes that make cells more likely to become cancerous and more difficult to treat. This review explains how oncogenes cause RS and how cells cope with it, leading to the development of cancer. Understanding these processes can help in developing new cancer treatments.

**Abstract:**

Activation of oncogenes disturbs a wide variety of cellular processes and induces physiological dysregulation of DNA replication, widely referred to as replication stress (RS). Oncogene-induced RS can cause replication forks to stall or collapse, thereby leading to DNA damage. While the DNA damage response (DDR) can provoke an anti-tumor barrier to prevent the development of cancer, a small subset of cells triggers replication stress tolerance (RST), allowing precancerous cells to survive, thereby promoting clonal expansion and genomic instability (GIN). Genomic instability (GIN) is a hallmark of cancer, driving genetic alterations ranging from nucleotide changes to aneuploidy. These alterations increase the probability of oncogenic events and create a heterogeneous cell population with an enhanced ability to evolve. This review explores how major oncogenes such as RAS, cyclin E, and MYC induce RS through diverse mechanisms. Additionally, we delve into the strategies employed by normal and cancer cells to tolerate RS and promote GIN. Understanding the intricate relationship between oncogene activation, RS, and GIN is crucial to better understand how cancer cells emerge and to develop potential cancer therapies that target these vulnerabilities.

## 1. Introduction

Maintaining accurate genetic information and genome stability is a major challenge in living organisms. Acute and deleterious stresses induced by chemicals, radiation, and viral infections often cause severe DNA damage and/or disruption of genetic information, leading to toxic consequences [1,2]. However, we have limited knowledge on how cells respond to and tolerate chronic stresses caused by oncogene activations, which promote precancerous cell viability and clonal expansion.

A substantial body of evidence indicates that genomic instability (GIN) could be a fundamental driving force behind the acquisition of hallmarks of cancer through the gain and/or loss of gene function derived from structural variations of genomic DNA and alternations in genetic and/or epigenetic codes, which are implicated in the development and progression of cancer [3,4]. In hereditary cancers, the presence of GIN has been linked to mutations in tumor suppressor genes such as DNA repair genes, strongly supporting the mutator hypothesis, which states that GIN is present in precancerous lesions and drives tumor development by increasing the spontaneous mutation rate [5]. In colorectal cancer (CRC), microsatellite instability (MSI), a hypermutable phenotype, is caused by the loss of DNA mismatch repair activity, which is detected in approximately 12% of sporadic CRC [6]. However, in various sporadic cancers, the frequency of the inactivation of DNA repair genes is limited; instead, the activation of oncogenes has been proposed to be attributed to GIN [7]. Oncogene activation during cell proliferation can significantly disrupt the intracellular environment, leading to alterations in metabolites, transcription, and chromatin state, frequently resulting in disruptions to normal DNA replication, a phenomenon broadly referred to as replication stress (RS). Oncogene-induced RS is increasingly recognized as an early driver of GIN in the initial step of cancer development [8,9], whereas the mechanisms by which it promotes GIN and how cells fine-tune oncogene-induced RS to prevent cell death have only recently begun to be understood. Recently, we proposed a model in which ATR kinase, a key regulator of RS, confers tolerance to oncogene-induced RS, enabling precancerous cells to survive and proliferate [10]. It is anticipated that acute proliferative signals induced by oncogene activation perturb cellular homeostasis [8], inducing RS, whereas GIN is accelerated in RS-tolerated cells [10] (Figure 1). This supports the evidence that a small subset of the cells accumulating GIN continues to grow as an origin of the cancer in seemingly noncancerous tissues, causing clonal expansion [11].

Here, we provide an overview of the impact of major oncogene activation on DNA replication and discuss the cause and consequence of oncogene-induced RS, including RS tolerance (RST) mechanisms. We then illustrate the molecular basis underlying GIN progression under oncogene-induced RS and its potential importance in the early stage of cancer development.

## 2. The Impact of Oncogene Activation on DNA Replication

For more than a decade, a number of studies have expanded our knowledge of the mechanisms underlying oncogene-induced RS leading to GIN. These studies have revealed the intricate molecular underpinnings of this process, in which oncogene activation disrupts the tightly controlled process of DNA replication, by (1) physical disturbances on the replication fork [12,13], (2) insufficient dNTPs supply limiting the activity of DNA polymerases [8,14], and (3) base-alternations in DNA by chemical modification (or combinations of (1)–(3)) [15]. In the following section, we focus primarily on the activation of well-known oncogenes, such as oncogenic RAS mutations, cyclin E dysregulation, and MYC amplification, and discuss their impact on DNA replication.

### 2.1. RAS-Induced RS

#### 2.1.1. The Function of Oncogenic RAS

The RAS family is composed of three genes (H-, K-, and N-RAS) and encodes a low-molecular-weight GTPases protein that acts as a molecular switch, cycling between an inactive guanosine diphosphate (GDP)-bound state and an active guanosine triphosphate (GTP)-bound state at the plasma membrane [16], regulating cellular proliferation and survival via the RAS-MAP kinase cascade in response to extracellular stimulations [17]. This cycle is mediated by two other classes of proteins. Guanine nucleotide exchange factors (GEFs) promote the activation of RAS by aiding in the exchange of GDP for GTP, whereas GTPase-activating proteins (GAPs) promote RAS-mediated GTP hydrolysis, resulting in inactivation of RAS [16]. The KRAS protein consists of two domains, termed the G-domain (guanine nucleotide-binding domain), which is associated with GTPase activity, and the hypervariable region domain (HVR domain), which is required for membrane localization. The P-loop region (10-17 AA), located in the G-domain, is highly conserved between paralogs and is necessary for the binding of GTP and GDP, as well as for interactions with GEFs and GAPs [18]. Since the GAPs stimulate the GTPase activity of KRAS 100,000-fold more than the GDP-bound “OFF” state [19], it is evident that the interaction of KRAS and GAPs via the P-loop is required to prevent constitutive activation of KRAS. Thus, oncogenic missense mutations in the P-loop region, particularly at residues G12 or G13, impair the interaction with GAPs or the catalysis of GTP hydrolysis, resulting in the chronic binding of GTP to KRAS and its consistent activation. The KRAS gene exhibits a high mutation rate among all cancers [20], and is particularly associated with 78.7% of pancreatic adenocarcinoma (PAAD), 20.1% of lung adenocarcinoma (LUAD), and 49.7% of colorectal adenocarcinoma (COAD) [21]. Mutations in the HRAS and NRAS have also been observed in other cancer types, including head and neck, bladder, and skin cancers [21].

#### 2.1.2. The Entity of Oncogenic-RAS Induced RS

How does oncogenic RAS affect DNA replication in the early stages of tumorigenesis? In recent years, DNA fiber assays have remarkably opened new avenues for research into the mechanisms of oncogene-induced RS [22]. In a landmark study, Di Micco and co-workers showed that retroviral introduction of HRAS^G12V^ induces RS in normal human fibroblasts (BJ cells) by increasing origin firing and generating asymmetric replication forks, and that oncogene-induced senescence (OIS) results from the enforcement of the DNA damage response (DDR) triggered by oncogene-induced RS [23]. The impact of oncogenic RAS on DNA replication was further reported in mouse embryonic fibroblasts (MEFs) following retroviral introduction of KRAS^G12V^. This leads to a reduction in fork speed and subsequent activation of the ATR pathway [24]. These observed phenotypes are strikingly similar to those observed in the osteosarcoma cell line, U2OS cells, overexpressing cyclin E [25]. Subsequently, it has been shown that doxycycline-inducible HRAS^G12V^ expression in human fibroblasts (BJ cells) slows fork speed, presumably as a consequence of oxidative stress caused by cellular metabolic changes including reactive oxygen species (ROS) [26]. Several studies have proposed a model in which oncogenic RAS induces DNA damage and OIS by generating ROS, which in some cases leads to nucleotide oxidation [27,28,29,30,31], but direct evidence as to whether it is necessary to slow down the fork speed has not been clarified. It has also been demonstrated that retroviral introduction of oncogenic HRAS^G12V^ in primary human normal fibroblasts (IMR90 cells) interferes with cellular dNTP levels by downregulating ribonucleotide reductase subunit M2 (RRM2), causing dNTP pool depletion and RS, as indicated by premature termination of replication forks [32] (Figure 2).

Kotsantis and colleagues have reported that increased transcription can be a mechanism of RAS-induced RS. Elevated RNA synthesis in BJ cells expressing tamoxifen-inducible HRAS^G12V^ causes replication fork slowing and DNA damage through transcription–replication conflict (TRC) and the subsequent formation of R-loops [9]. TRCs occur when the transcription and replication machineries collide on the same DNA template. These collisions can lead to the formation of R-loops, which are stable RNA–DNA hybrids that can displace the non-template DNA strand. TRCs impede replication fork progression as a result of head-on or codirectional collisions between the two machineries or between replication and R-loops, causing replication stress [12]. RAS-induced enhanced transcription is driven by the upregulation of transcription factors such as TBP (TATA-box binding protein) [33]. Interestingly, overexpression of TBP alone causes RS, DNA damage, and senescence, suggesting that oncogenic RAS-induced elevated RNA synthesis and/or subsequent TRC and R-loop formation in normal human fibroblasts may be sufficient to induce RS, which promotes GIN [34] (Figure 2).

Collectively, these studies indicate not only that oncogenic activation of KRAS or HRAS in fibroblasts, regardless of their induction mechanisms and species background, leads to RS by the multiple mechanisms mentioned above and others [8], but also that it triggers the DDR, followed by cell death or senescence, which functions as an inducible barrier against progressive GIN [23,25,35,36]. However, this raises the question of how oncogenic RAS-expressing cells adapt to RS while driving GIN, despite the availability of the ATR/ATM-mediated tumorigenesis barrier.

#### 2.1.3. The Consequence of RAS-Induced RS During Gaining of GIN

A clue for considering biomolecular regulators adapting oncogenic RAS-induced RS resides in ATR signaling, which responds to a broad spectrum of RS. Previous studies have reported that heterozygotes for the ATR gene, which reduce ATR protein expression to about half, are tumor-prone because of partial DDR defects leading to GIN that accelerate the incidence of lung adenocarcinoma induced by oncogenic KRAS^G12D^ [37]. Notably, a marked decrease in ATR levels (approximately 1/10th of normal levels) due to hypomorphic mutations in ATR mimicking Seckel syndrome induces a dramatic increase in DNA damage and suppresses tumor growth in cells experiencing suprathreshold RS induced by HRAS^G12V^ [38]. Therefore, a certain level of ATR expression appears to be necessary to reduce DNA damage to non-lethal levels and allow tumor cell growth under oncogenic RAS-induced RS. Furthermore, an extra allele of Chk1 protects mouse fibroblasts from RS and enhances HRAS^G12V^-induced transformation by reducing DNA damage-associated apoptosis [39]. Similarly, overexpression of Claspin and Timeless, which consist of a functional module with Chk1, promotes fork progression and protects cells from HRAS^G12V^-induced RS in a DDR-independent manner [40]. In these circumstances, Chk1 activation can be increased by overexpressing Claspin, leading to repriming activation on the leading-strand template due to the activation of PrimPol, which facilitates the completion of DNA synthesis when replication forks encounter obstacles that cause polymerase stalling [41]. A recent study showed that Timeless is essential for the engagement of PARP1 to the replisome to coordinate lagging strand synthesis with replication fork progression [42]. Moreover, PARP1 functions together with Timeless and Tipin protect the replisome from TRCs in the early S phase [43]. These studies thoughtfully suggest that upregulation of the ATR-Chk1 axis is efficient for tolerance to oncogenic RAS-induced RS, while suppressing DDR-mediated cell death, which is a barrier to tumorigenesis, highlighting the dosage-dependent dual function of ATR modules, on one hand as a barrier for tumor development and on the other hand as a supporter of cell survival in response to oncogenic-RAS activation during cancer progression (Figure 1).

Recently, we reported that KRAS^G12V^ induces RS by decreasing replication fork speed in normal human lung epithelial cells (SAEC) [10]. Interestingly, the established RS-tolerant cells (RSTCs) under KRAS^G12V^-induced RS express increased levels of ATR compared to normal cells and recover their fork speed. Furthermore, normal lung epithelial cells overexpressing ATR, mimicking RSTC, also showed unrestrained fork progression in the presence of KRAS^G12V^ expression, suggesting that increased ATR protein expression is required for RST to ensure replication fork progression under KRAS^G12V^-induced RS. Despite seemingly normal fork progression in RSTCs contributing to complete genome duplication, PrimPol-repriming regulated by ATR/Chk1 kinase leads to the accumulation of ssDNA gaps in nascent DNA, increasing the risk of spontaneous double-strand break (DSB) leading to GIN. These findings raise the previously unrecognized possibility that ATR-PrimPol plays a role in enabling cells to complete DNA replication and survive under oncogenic KRAS-induced RS, in return for which cells accumulate genomic alterations and expand GIN, a driving force for cancer development and malignancy [10]. Consistently, elevated levels of KRAS, ATR, and Chk1 correlate with the proliferative potential of tumor cells, aneuploidy and the appearance of metastasis in endometrial cancer patients with relapse, indicating that activation of the ATR-Chk1 axis in cooperation with KRAS expression, also increases the risk of recurrence [44]. Alternatively, in response to HRAS^G12V^ induction, human foreskin fibroblasts trigger RST by increasing Topoisomerase 1 (TOP1) expression to deal with the cause of RS, namely the TRC-associated R-loop, resulting in the acceleration of replication fork speed and exacerbation of DNA damage and GIN [45]. Together, RST-dependent GIN progression may provide new insights into oncogenic RAS-driven tumor development or its recurrence.

#### 2.1.4. Exploring Causes of Oncogenic RAS-Induced RS

One of unexpected finding of our study is that, consistent with previous reports, RNA transcription causes reduced replication fork progression, whereas the resulting R-loops and TRCs are not critical determinants of the reduction in fork progression induced by KRAS^G12V^ in lung epithelial cells [10]. Instead, KRAS^G12V^ induces unscheduled transcription followed by polycomb repressive complex 2 (PRC2)-mediated chromatin remodeling via H3K27me3, leading to locally compacted facultative heterochromatin formation, which may be an obstacle to DNA replication (Figure 2). The idea that local heterochromatin formation during DNA replication may be a cause of RS is increasingly recognized [46,47,48] and highlights the importance of considering chromatin regulation in how oncogenes or exogenous stress induces RS. Furthermore, pharmacological suppression of H3K27me3 methyltransferase, such as EZH2, impairs oncogenic KRAS-driven lung tumor growth in vivo [49], suggesting that chronic RS induced by heterochromatin and its tolerance may further drive the progression of GIN to promote its malignancy. However, it remains unclear how oncogenic RAS induces RS via the TRC-associated R-loop or transcription-associated heterochromatin, and why they are altered in a cell lineage-dependent manner. Although oncogenic HRAS and KRAS mutations are most frequently found in epithelial cell-derived adenocarcinomas, their impacts have been discussed primarily on the basis of the fibroblast phenotype in the field of aging and cancer biology. Perhaps, this discrepancy may stem from the historical background that the oncogenic potential of the HRAS and KRAS genes was validated by transducing them specifically into murine fibroblast NIH3T3 cells [50,51,52,53]. Given the significant differences between fibroblast and epithelial cell environments, it is essential to proceed with caution when extending the conclusions drawn from fibroblast-based RS and GIN studies to the intricate mechanisms of epithelial cell tumorigenesis.

### 2.2. Cyclin E-Induced RS

#### 2.2.1. The Function of Cyclin E

Cyclin E is a pivotal protein in the intricate machinery governing the cell cycle. The cyclin E family is composed of two proteins: cyclin E1 and E2 (CCNE1 and CCNE2). As a core component of the cyclin-dependent kinase (CDK) complex, specifically with CDK2, it orchestrates the transition from the G1 to S phase, a critical step in DNA replication [54]. In the G1 phase, the cyclin E/CDK2 complex phosphorylates and promotes degradation of the cell cycle regulating factor the retinoblastoma protein (RB), leading to the release of the E2F transcription factor [55,56]. E2F-mediated cyclin E transcription leads to cyclin E protein accumulation, which peaks at the G1/S transition. The cyclin E/CDK2 complex then phosphorylates numerous substrates, controlling essential cellular processes including progression through the restriction point and initiation of DNA replication. By the end of the S phase, cyclin E is completely degraded by ubiquitin-mediated proteolysis, eliminating cyclin E/CDK2 activity until the subsequent G1 phase [57,58]. Aberrant cyclin E expression or function, which results from CCNE1 amplification, overexpression, or impaired protein degradation, is frequently observed in various cancer types such as ovarian, breast, lung tumors, and leukemias [57,59,60,61].

#### 2.2.2. The Entity of Cyclin E-Induced RS

Cells with high cyclin E expression are affected by chronic RS. In an earlier study, it was shown that overexpression of cyclin E introduced by the adenovirus system in human nasopharyngeal epidermoid carcinoma cells interferes with the assembly of the components of the pre-replication complex MCM2, MCM4 and MCM7 into chromatin during late mitosis and early G1 phase [62], resulting in the abrogation of origin licensing. In this state, a smaller number of origins fire and the rate of DNA synthesis decreases around each replication origin (Figure 3) [20,62]. Subsequently, retrovirally overexpressed cyclin E was shown to impair replication fork progression in U2OS cells [26]. In BJ cells, retrovirally overexpressed cyclin E promoted cell proliferation with insufficient nucleotide levels, resulting in reduced fork progression, which was rescued by exogenous nucleoside supplementation (Figure 3) [63].

Another predominant mechanism for cyclin E-induced RS is interference between replication and transcription. Impaired replication fork progression induced by the overexpression of cyclin E (tetracycline-repressive system) in U2OS cells is recovered by inhibiting replication initiation factors, indicating that excessive origin firing causes replication slowing [64]. In the same study, inhibition of transcriptional elongation was shown to alleviate replication stress and reduce DNA damage caused by cyclin E1 overexpression [64]. These results indicate that RS in cyclin E-overexpressing cells likely results from increased replication initiation coupled with conflicts between replication and transcription. This model is further supported by a recent study in which firing of intragenic origins caused by premature S phase entry represents a mechanism of overexpression of cyclin E-induced DNA replication stress [65]. In yeast, the length of G1 is sufficient for transcription to inactivate origins across the entire length of genes [66]. However, oncogenes greatly reduce the length of G1 [67], and therefore leave insufficient time for transcription to inactivate all intragenic origins (Figure 3). These excessive and ectopic origin firings lead to increased TRCs and R-loops [64,65], which are the major cause of RS induced by cyclin E [68,69].

#### 2.2.3. The Consequence of Cyclin E-Induced RS During Gaining of GIN

Deregulated cyclin E-induced RS causes DNA damage which activates checkpoint responses that regulate anti-tumor barriers (such as cell growth arrest, senescence, and cell death) and GIN. The ATR/ATM-regulated DNA damage response is activated promptly in a time-dependent manner (2 to 6 days) after the induction of cyclin E in U2OS cells and human fibroblasts [25,70]. Consistent with the observation in cultured cells, these activated DNA damage responses are commonly detected in human tumor tissue from different stages and early precursor lesions (but not normal tissues) [36,70]. Importantly, DDR activation precedes GIN, represented by the occurrence of p53 mutations and/or defects in DNA damage signaling [36,70], indicating that early in tumorigenesis (before GIN), human cells activate an ATR/ATM-regulated DNA damage response network that delays or prevents cancer. However, constitutively overexpressing cyclin E (during a 30-day growth period) in rat embryo fibroblasts and human breast epithelial cells results in an increased proportion of aneuploid cells, indicative of chromosome instability (CIN) [71]. These data indicate that changes in cyclin E/Cdk2 kinase activity may affect processes involved in faithful chromosome replication and segregation.

How do cells overexpressing cyclin E overcome the barrier for GIN and progress to cancer cells that have acquired tolerances for the RS? Recently, it was reported that F-box/LRR-repeat protein 12 (FBXL12), a substrate recognition component of the SCF (SKP1-CUL1-F-box protein)-type E3 ubiquitin ligase complex, protects replication forks under cyclin E-induced RS. Chk1-mediated phosphorylation of FANCD2 triggers its FBXL12-dependent degradation and promotes efficient DNA replication. High FBXL12 expression is associated with poor survival in patients with high CCNE1 expression, indicating that FBX12-dependent RST initiated by ATR-Chk1 axis activation may contribute to cyclin E-driven tumor malignancy [72]. Alternatively, cyclin E overexpression in RPE cells causes the transmission of DNA lesions into mitosis, which triggers RAD52-dependent mitotic DNA synthesis (MiDAS), supporting cell survival. Moreover, cyclin E1 amplification is associated with increased RAD52 expression in breast cancers, suggesting that Rad52 mediates RST [73].

Although aneuploidy in cancer genomes strongly correlates with mutations in TP53 [74], Zeng and colleagues reported that cyclin E-induced RS drives p53-dependent whole-genome duplication, an important driver of aneuploidy [75]. Cyclin E-induced RS in U2OS and RPE cells prolongs G2 phase arrest in an ATR/Chk1 checkpoint-dependent manner. p53, through its downstream target p21, whose accumulation is also dependent upon checkpoint activation, together with Wee1, sufficiently inhibits mitotic CDK activity to activate APC/C-Cdh1 and promote mitotic bypass [75]. Cyclin E expression prevents cells entry into senescence, continues to progress cell cycle and can drive senescent cells to complete endoreduplication, resulting in cells proliferation while acquiring GIN [75].

### 2.3. MYC-Induced RS

#### 2.3.1. The Function of MYC

The MYC family of oncogenes contains three well-defined members: MYC (also known as c-Myc), MYCN, and MYCL, which encode transcription factors that control gene expression involved in cell proliferation and differentiation, thereby contributing to tumorigenesis and reprogramming [76]. c-Myc is frequently overexpressed and/or activated in a wide variety of cancer types, while its two paralogs, N-Myc and L-Myc are associated with neuroblastoma and small cell lung cancer, respectively. MYC deregulation occurs through genetic alterations, including amplification and translocation, aberrant signal transduction leading to the increased MYC expression, or MYC protein stabilization [77,78]. Pancancer analysis across the 33 cancers of The Cancer Genome Atlas (TCGA) identified focal amplification (28% of the samples) in at least one of the three MYC families, and MYC antagonists were mutated (MGA, 4% of samples) or deleted (MNT, 10% of samples) [79]. Hence, MYC activation has been implicated in the initiation, progression, and maintenance of most types of cancers [80]. MYC largely functions as a transcription factor that coordinates many biological processes associated with the features of cancer, including autonomous proliferation and growth, increased protein biogenesis, and global changes in cellular metabolism. Accumulating evidence shows that the MYC family is also a major driver of oncogene-induced RS, as described below.

#### 2.3.2. The Entity of MYC-Induced RS

MYC activation results in dramatic changes in the transcriptional program. MYC acts as a universal amplifier of expressed genes by directly targeting the promoter regions of active genes and producing increased levels of transcripts within the cell’s gene expression program [81,82], providing an explanation for the diverse effects of oncogenic MYC on gene expression in different tumor cells. In contrast, MYC activates and represses the transcription of discrete gene sets, leading to indirect transcriptional amplification via feedback on global RNA production and turnover [83,84]. These MYC-dependent transcriptional activations contribute to shaping the gene expression profiles that are essential for tumor initiation and growth. Importantly, increased RNA synthesis causes the R-loop and TRC, which can cause transcription-associated RS [9,85,86]. Recent studies have shown that c-Myc expression is associated with R-loop accumulation in breast cancer cells [87,88], thereby conferring a synthetic lethality by inhibiting TOP1 [88], an enzyme that relaxes DNA supercoiling and prevents R-loop formation [89,90]. These observations suggest that MYC potentially induces transcription-associated RS, most likely through R-loop formations followed by TRCs (Figure 4). However, given recent study showing that inhibition of RNA polymerase does not rescue replication fork stalling induced by c-Myc in RPE1 cells [91], direct evidence that MYC-dependent transcription impairs replication fork progression has not yet been well established.

By contrast, MYC regulates several mechanisms to prevent transcription-associated RS. MYC has been shown to physically interact with TOP1/2 and stimulate their activity in a “topoisome”, suggesting that their enzymatic activities may resolve DNA topological stress during transcription-associated RS [92]. In addition, MYC multimers, often sphere-like structures, are formed in response to RS and accumulate on chromatin immediately adjacent to stalled replication forks and surround the FANCD2, ATR, and BRCA1 proteins, which limits DNA damage in the S phase [93]. In neuroblastoma cells, N-Myc-dependent recruitment of BRCA1 and USP11 to transcriptional pause sites facilitates mRNA de-capping and the release of stalled RNA polymerases, thereby suppressing R-loop accumulation [94]. N-Myc also interacts with and activates Aurora A on chromatin, which phosphorylates histone H3 at serine 10 in the S phase, promotes the deposition of histone H3.3 and suppresses R-loop accumulation and TRC [95,96]. Therefore, these results indicate that MYC family proteins play a role to resolve transcription-mediated problems on DNA, possibly reducing their contribution as a cause of replication stress induced by MYC activation. Since MYC-driven cancer cells are vulnerable to inhibition of Aurora A and ATR which suppress increases in R-loop, TRC, and DNA damage [96], MYC activation may not only cause transcription-associated RS but also elevate the tolerability for survival and growth under this stress.

#### 2.3.3. Non-Transcriptional Role of MYC in RS

The role of MYC in cell cycle progression is generally linked to its transcriptional regulation of cyclins and CDKs. Since cyclin E overexpression markedly induces RS, as described above, dysregulation of cell cycle progression may be a cause of MYC-induced RS. On the other hand, MYC is directly involved in the regulation of DNA replication in a non-transcriptional manner [97,98,99]. Notably, Dominguez-Sola and colleagues revealed that c-Myc interacts with the pre-replicative complex and is required for efficient DNA replication and proper origin specification in the absence of transcription [100]. In addition, c-Myc induces the decondensation of higher-order chromatin at targeted sites, promoting Cdc45/GINS recruitment to resident MCMs and the activation of CMGs [101]. Consistent with these reports, c-Myc overexpression alters the spatiotemporal program of replication initiation by increasing the density of early replicating origins, resulting in replication fork stalling/collapse and subsequent DNA damage [100,102] (Figure 4). c-Myc overexpression increases cohesins chromatin occupancy at CTCF sites, interfering with the progression of replication forks and contributing to c-Myc-induced RS [91] (Figure 4). Together, these results indicate that MYC-driven cancer cells are exposed to diverse causes of RS owing to their multifunctional roles.

#### 2.3.4. MYC-Mediated RST Mechanisms During Gaining of GIN

As a result of RS, MYC activation leads to DNA damage through direct impairment of DNA replication dynamics [100,101], followed by DDR activation, which potentially limits cell proliferation and survival [103]. Nevertheless, MYC is also involved in diverse RST mechanisms. MYC binds to the promoters of DNA repair genes such as Rad51, and endogenous MYC protein expression correlates with RAD51 protein expression [104]. The MRN complex is transcriptionally upregulated by c-Myc [105] and N-Myc [106], and suppresses N-Myc-mediated RS and DNA damage to support tumor growth [107]. In fibroblasts, c-Myc directly stimulates the transcription of WRN helicase, whose depletion results in increased RS levels and senescence induction [108,109]. Importantly, WRN depletion impaired tumor growth in c-Myc-driven non-small cell lung cancer xenografts and Eμ-Myc-driven B-cell lymphoma in a mouse model, suggesting that WRN upregulation by c-Myc promotes tumorigenesis [110]. Thus, MYC, in a transcription-dependent manner, induces DDR factors associated with DNA repair and their functions contribute to the establishment of a safeguard system against high RS levels induced by MYC itself.

Furthermore, MYC-driven cancer cells depend on various context-dependent RST mechanisms. MYC-driven cancer cells, in which RS is limited and tumor growth is supported by ATR/Chk1 activation, exhibit higher sensitivity to ATR and Chk1 inhibitors [38,111,112,113]. In mouse models of Eμ-Myc-driven B-cell lymphoma, the DNA translocases SMARCAL1 and ZRANB3 protect replication forks [114]. In U2OS cell models, depletion of Polη, a Y-family TLS polymerase, enhanced c-Myc-induced replication fork stalling and subsequent DNA damage, indicating that Polη mediates RST under c-Myc-induced RS [115]. Furthermore, nucleotide depletion and subsequent RS and DNA damage induced by HPV E6/E7 and cyclin E oncogenes are rescued by c-Myc expression-dependent increased transcription of nucleotide biosynthesis genes [63]. Similarly, oncogenic RAS-induced DNA damage can be rescued by exogenously supplied nucleotides [26]. Considering the interdependency between RAS, cyclin E, and MYC, MYC upregulation, in a feedback manner, may contribute to mitigating RAS- and cyclin E-induced RS, generate genome instability at tolerable levels, and promote cancer development.

## 3. Conclusions

The interplay between oncogenes, RS, and GIN emerges as a critical factor in cancer development. Oncogene activation triggers RS by disrupting the DNA replication processes, leading to replication fork stalling, collapse, and DNA damage. While the DDR or OIS initially acts as a barrier, a subset of cells develop RST mechanisms presumably in an ATR/Chk1 signaling pathway-dependent manner, enabling them to survive or escape senescence, and proliferate. Alternatively, DNA damage induced by RS and persisting during mitosis may be repaired through less accurate repair pathways, such as single-strand annealing (SSA) or alternative end-joining/microhomology-mediated end-joining (alt-EJ/MMEJ), promoted by RAD52 and the mutagenic DNA polymerase Pol θ, respectively [116]. Mitotic MMEJ helps prevent cell division from proceeding with unrepaired DNA damage, thereby contributing to ensuring cell viability [117]. These survival advantages allow for clonal expansion and the accumulation of genetic alterations, contributing to GIN, a hallmark of cancer (Figure 1). In addition, several studies have proposed a model in which a subset of cells undergoing OIS acquire a mechanism for re-entry into the cell cycle [118,119], which would promote tumor development [120]. Collectively, the tolerance or escape of genotoxic stress induced by oncogene activation while gaining GIN and expanding effectively is a crucial matter for cells during the initial step of tumorigenesis.

However, various oncogenes may have different effects on the RS and RST pathways depending on the cell type and the time point at which their effects are addressed, such as before or after cell transformations, which also applies to GIN. Oncogenic KRAS-induced RST cells exhibit COSMIC signature SBS8, which likely arises from uncorrected late replication errors [10,121]. In the case of the CDC6 oncogene, early acquired recurrent chromosomal inversions have been reported at the locus encoding the circadian transcription factor BHLHE40 [119]. Thus, it is difficult to predict whether a specific type of GIN will be induced by oncogene-induced RS, making this a promising area for future research. Furthermore, the cause of RS may be spontaneously and stochastically induced by unscheduled transcription, dysregulated chromatin organization, and perturbed cell cycles despite the absence of oncogene activations, at an unrecognizably low level in normal cells, leading to early genome alterations called “bad luck” [122], conceivably through RST according to the specific circumstances. Indeed, recent high-throughput readouts of the human genome have demonstrated that GIN arises in noncancerous cells in correlation with aging and environmental risks [123]. Therefore, further research elucidating the specific molecular mechanisms underlying RST in different contexts and how GIN arises and evolves in the early or late stages of tumorigenesis could lead to insights into the identification of vulnerabilities associated with RS and GIN for the development of novel cancer therapeutic approaches and potential preventive cancer strategies.

## Figures and Tables

**Figure 1 cancers-16-03507-f001:**
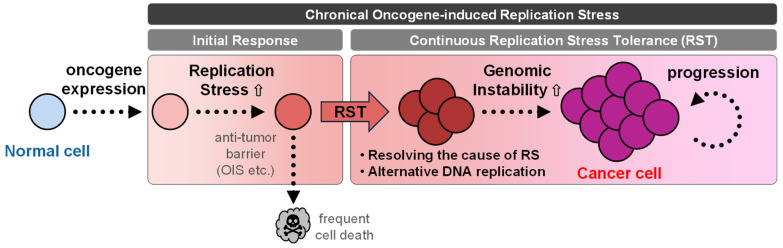
The replication stress tolerance process and cancer progression. Oncogene activation dysregulates the DNA replication via modifying metabolisms, the cell cycle, and the replicative environment, leading to replication stress (RS). During the initial response to oncogene-induced RS, an “anti-tumor barrier”, such as DDR activation and/or OIS, is triggered. A small subset of the cells that acquire replication stress tolerance (RST) mechanisms by resolving the cause of RS and/or altering the DNA replication process begins to clonally expand, while RST contributes to genomic instability (GIN).

**Figure 2 cancers-16-03507-f002:**
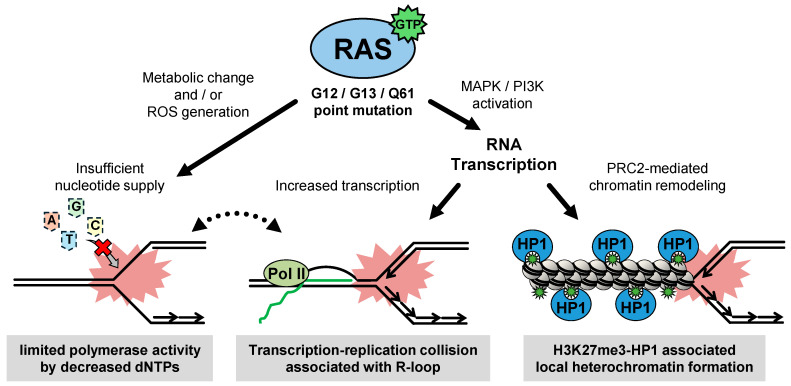
Types and causes of RAS-induced replication stress. RAS-mediated metabolic alternation reduces the nucleotide supply, attenuating DNA polymerase catalysis and leading to replication stress (RS). RAS also stimulates the signal cascade, increasing the amount of global RNA synthesis and inducing transcription replication collision (TRC) associated with R-loop. Alternatively, feedback to alternations in RNA synthesis results in local chromatin compaction, interfering with DNA replication.

**Figure 3 cancers-16-03507-f003:**
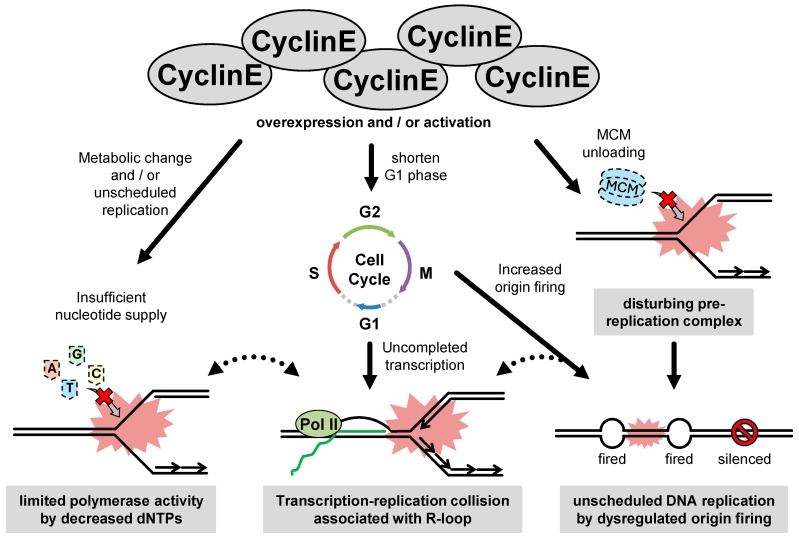
Types and causes of excessive cyclin E-induced replication stress. Overexpressed cyclin E reduces the nucleotide supply and hinders nascent DNA synthesis. Dysregulated cyclin E specifically shortens the length of the G1 phase. Entry to the premature S phase induces uncompleted transcription and ectopic origin firing, resulting transcription replication collision (TRC). Cyclin E compromises the regulation of origin firing. Cyclin E can disrupt the normal control of DNA replication initiation sites, leading to unscheduled replication and increased TRC, leading to replication stress (RS).

**Figure 4 cancers-16-03507-f004:**
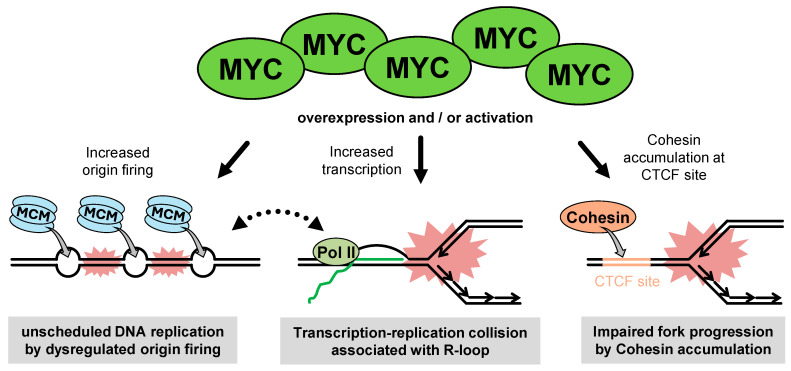
Types and causes of oncogenic MYC-induced replication stress. MYC overexpression and/or activation induces replication stress (RS) through both transcriptional and non-transcriptional mechanisms. MYC-dependent transcriptional activation causes R-loop accumulation followed by transcription–replication collisions. In non-transcriptional mechanisms, MYC overexpression promotes Cdc45-MCM-GINS activation leading to unscheduled origin firing with subsequent replication fork stalling and DNA damage. In addition, MYC overexpression increases cohesion accumulation on chromatin in a CTCF-dependent manner, interfering with replication fork progression.

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
