# Peer review of "Tolerance of Oncogene-Induced Replication Stress: A Fuel for Genomic Instability"

_cancers, 2024, doi:10.3390/cancers16203507_

Round 1

Reviewer 1 Report

Comments and Suggestions for Authors

This is a timely review in a field which is rapidly expanding and this manuscript entitles “Tolerance of oncogene-induced replication stress: a fuel for genomic instability” by Taichi Igarashi et al. focuses on oncogene induced-RS which leads to replication forks stalling or collapse, and DNA damage. The authors provide an overview of the impact of major oncogene activation, such RAS, Cyclin E and MYC, on DNA replication and address the cause and consequence of oncogene-induced RS, notably the activation of RS tolerance (RST) mechanisms as well as the processes underlying genomic instability under oncogene-induced RS and their importance in the early stage of cancer development.

Overall, the review is informative and provides useful background on the induction of RS by some major oncogenes and how cells cope with it and tolerate RS, leading to genomic instability and cancer.

I have just one comment, the DNA repair mechanisms such SSA (RAD52) or TMEJ/MMEJ (Polq) are never mentioned as processes that could also play a role in tolerating RS, do these pathways never contribute to cell survival to oncogene-dependent RS?

After answering this question and incorporating eventually this potential missing mechanism, the review would be suitable for publication in cancers. 

Comments on the Quality of English Language

Minor editing of English language is required

Reviewer 2 Report

Comments and Suggestions for Authors

Igarashi et al. discuss mechanisms of genome instability induced by certain oncogene expression. This review highlights some of their recent published work regarding Ras, and this discussion is appropriate context for considering the effects of other oncogenes.

Overall comments

Can we learn anything about oncogene induced replication stress by looking at specific types of genome instability that are induced? Is there a unique signature for particular oncogene-induced replication stress?

Given the importance of replication/transcription conflicts, I would like to see a slightly more in-depth discussion of this issue.

Specific comments

Lines 54-56 “In sporadic 54 cancer, by contrast, frequency of inactivation of DNA repair genes is limited and, instead, activation of oncogenes has been proposed to attribute to GIN.” I am not sure I agree; it would be useful to provide references that suggest that DNA repair deficiency frequently occurs even in sporadic cancers. Does mismatch repair count as a DNA repair pathway?

Lines 56-59 “Complex environmental changes during cell growth driven by the oncogene activations often lead to disturbances in physiological DNA replication, which are widely referred to as replication stress (RS)”. I think “complex environmental changes” is a nebulous concept of limited usefulness.

Line 174 “Claspin” and “timeless” need a bit more description.

Line 248 …perpetual replication stress”. Explain and reference this concept please.

Line 375 …Myc cells are sensitive to Top1 inhibition… so are just about all proliferating cells (and note by Top1 inhibition, the authors mean treatment with Top1 poisons such as camptothecin. I think this argument is rather weak.

Minor points

Lines 44-46 Rewrite for correct grammar

Comments on the Quality of English Language

The English is understandable, but there are quite a few minor grammatical errors. I pointed out one specific one in my comments to authors. The paper would be improved by editing for ENglish usage.
